# Photocatalytic Performance of Sol-Gel Prepared TiO_2_ Thin Films Annealed at Various Temperatures

**DOI:** 10.3390/ma16155494

**Published:** 2023-08-07

**Authors:** Lu He, Dietrich R. T. Zahn, Teresa I. Madeira

**Affiliations:** 1Semiconductor Physics, Chemnitz University of Technology, 09107 Chemnitz, Germany; zahn@physik.tu-chemnitz.de (D.R.T.Z.); teresa.madeira@physik.tu-chemnitz.de (T.I.M.); 2Center for Materials, Architectures, and Integration of Nanomembranes (MAIN), Chemnitz University of Technology, 09107 Chemnitz, Germany

**Keywords:** TiO_2_ thin film, sol-gel process, photocatalysis, thin film characterization

## Abstract

Titanium dioxide (TiO_2_) in the form of thin films has attracted enormous attention for photocatalysis. It combines the fundamental properties of TiO_2_ as a large bandgap semiconductor with the advantage of thin films, making it competitive with TiO_2_ powders for recycling and maintenance in photocatalytic applications. There are many aspects affecting the photocatalytic performance of thin film structures, such as the nanocrystalline size, surface morphology, and phase composition. However, the quantification of each influencing aspect needs to be better studied and correlated. Here, we prepared a series of TiO_2_ thin films using a sol-gel process and spin-coated on p-type, (100)-oriented silicon substrates with a native oxide layer. The as-deposited TiO_2_ thin films were then annealed at different temperatures from 400 °C to 800 °C for 3 h in an ambient atmosphere. This sample synthesis provided systemic parameter variation regarding the aspects mentioned above. To characterize thin films, several techniques were used. Spectroscopic ellipsometry (SE) was employed for the investigation of the film thickness and the optical properties. The results revealed that an increasing annealing temperature reduced the film thickness with an increase in the refractive index. Atomic force microscopy (AFM) was utilized to examine the surface morphology, revealing an increased surface roughness and grain sizes. X-ray diffractometry (XRD) and UV-Raman spectroscopy were used to study the phase composition and crystallite size. The annealing process initially led to the formation of pure anatase, followed by a transformation from anatase to rutile as the annealing temperature increased. An overall enhancement in crystallinity was also observed. The photocatalytic properties of the thin films were tested using the photocatalytic decomposition of acetone gas in a home-built solid (photocatalyst)–gas (reactant) reactor. The composition of the gas mixture in the reaction chamber was monitored using in situ Fourier transform infrared spectroscopy. Finally, all of the structural and spectroscopic characteristics of the TiO_2_ thin films were quantified and correlated with their photocatalytic properties using a correlation matrix. This provided a good overview of which film properties affect the photocatalytic efficiency the most.

## 1. Introduction

Titanium dioxide (TiO_2_) is a widely studied metal oxide semiconductor due to its diverse applications in the fields of electro-optical devices [1], self-cleaning [2,3,4], functional coatings including anti-fogging surface coatings [5,6,7] and corrosive coatings [8], solar cells [9,10,11], photocatalysis [12,13,14], various sensors [15,16,17,18], batteries [19,20,21,22], and energy harvesting [23,24,25]. TiO_2_ shows advantages over other photosensitive materials because of its high photocorrosion resistance in aqueous media, cost-effectiveness, ease of access, and environmental-friendliness [26]. TiO_2_ powder, such as the commercially available P25 powder from Degussa, has been studied thoroughly [27,28,29,30,31]. Such powders are widely used in aqueous environments as a mobile phase [32,33,34,35,36] to decompose contaminants. However, such processes usually require long reaction periods. Also, photocatalyst powders in aqueous environments need to be recycled afterwards [37]. Therefore, photocatalytic reactions using immobilized solids have become crucial for many application conditions [38,39,40,41]. Besides TiO_2_ [42,43,44,45], thin film-based photocatalysis using other materials has also been widely studied [46,47,48].

For photocatalytic particles that are immobilized in thin films, photo-induced electron–hole pairs are created and transferred to the boundary of the photocatalyst. The charges then interact with molecules adsorbed on the photocatalyst surface, initiating the photocatalytic reactions [49,50]. The optical and electrical properties depend on the nanostructures [51], phase composition [52], surface morphology [53], and doping [54,55,56,57,58,59]. Many methods can be used to prepare thin films, such as atomic layer deposition [60,61], pulsed laser deposition [62], chemical vapor deposition [63], radio frequency magnetron sputtering [64,65], as well as spray pyrolysis [66] and spin coating. The choice of one method over the other usually depends on the type of film structure targeted and the cost of processing.

TiO_2_ exists mainly in three different crystalline structures: anatase (trigonal), rutile (tetragonal), and brookite (orthorhombic) [67]. Their bulk phases have optical bandgaps slightly above 3 eV, for instance, 3.0 (I)–3.4(D) eV for rutile [68,69], 3.1 (I)–3.6(D) eV and 4.45 eV(D) for brookite [70], and 3.2 (I)–3.6(D) eV for anatase [71,72,73,74]. The anatase phase is metastable, while rutile is thermally more stable than anatase in general [75]. For nanocrystalline sizes below 15 nm, however, anatase becomes more stable than rutile [76,77]. Due to a trade-off between its optical absorption capability and relatively large redox potential for many chemical reactions, anatase and rutile are the most frequently reported phases used for photocatalytic applications [78]. Even though most reports state that anatase performs better than rutile [79,80], recent experimental and theoretical studies have demonstrated that a mixture of anatase and rutile in the right proportions should provide even better photocatalytic activity due to the fact that, in the mixed titania phase, a small rutile crystallite interweaves with anatase crystallites, thus allowing rapid electron transfer from rutile to anatase, thus creating enhanced catalytic spots at the anatase–rutile interfaces [81]. Studies focused on band alignment show that the photogenerated electrons could be more efficiently separated between the two phases, thus improving the photocatalytic performance [82,83,84]. These different perspectives found in the literature encouraged us to study and understand the phase in which TiO_2_ thin films are produced according to our procedure’s results in terms of the best photocatalytic activity.

In our study, we utilized a sol-gel process to prepare the thin film precursors, which were then spin-coated. The spin-coating technique is straightforward, time effective, simple to realize, cost-effective, offers broad versatility [85,86], and is currently becoming very popular in industrial applications, in which the spin coating of large surface areas is of interest [85]. The annealing temperatures were chosen in a vast range from 400 °C to 800 °C to cover the whole crystal phase formation and transformation from anatase, to anatase/rutile mixtures, to pure rutile. The step size of the annealing temperatures was as small as 20 °C, especially in the range where anatase and rutile coexist.

The samples were then characterized using various methods. Spectroscopic ellipsometry (SE) was used to determine the optical constants and film thickness. X-ray diffractometry (XRD) and ultraviolet (UV) Raman spectroscopy with an excitation wavelength of 325 nm were used to determine the phase and crystallite size. Atomic force microscopy (AFM) was used to determine the surface roughness and grain size. Finally, the photocatalytic properties of the TiO_2_ thin films were assessed using the photodegradation of acetone gas to carbon dioxide. Using a home-built photocatalytic chamber, the reaction was monitored via in situ Fourier transform infrared spectroscopy (FTIR).

All parameters influencing the photocatalytic performance were quantified and related to each other using a correlation matrix. This matrix provided an overview of the annealing temperature’s effect on the photocatalytic performance of titanium dioxide.

## 2. Sample Preparation

TiO_2_ thin films were prepared using a sol-gel process. We used a published recipe to prepare the precursor using a mixture of titanium tetraisopropoxide (TTIP) 97% from Sigma-Aldrich^®^ (Sigma-Aldrich Chemie GmbH, Taufkirchen, Germany), diethanolamine (DEA) from Sigma-Aldrich^®^, and ethanol (99.99%) with a volume ratio of 1:5:0.3 [87,88]. Boron-doped silicon (100)-oriented wafer pieces with a resistivity of 10 to 20 Ω·cm were used as substrates. The Si wafers were cut into pieces with square shapes and dimensions of about 1.5 × 1.5 cm^2^. The silicon substrates were then cleaned in an ultrasonic bath with acetone, ethanol, and deionized water for 15 min each.

The precursor was spin-coated using a two-step spin-coating process by a Laurell WS-650NZ-23NPP/LITE spin coater (Laurell Technologies, Lansdale, PA, USA). In the first step, a rotation speed of 2000 rotations per minute (rpm) was applied for 20 s, followed by the second step with a rotation speed of 3000 rpm for 10 s to obtain uniform spin-coated films [89,90]. For individual sample preparation, 70 µL of a precursor solution was first dropped onto the substrate to ensure that the substrates were fully covered before starting the spin coating.

All as-prepared samples were pre-baked on a hot plate in air at 200 °C immediately after the spin-coating process for one hour to promote solvent evaporation [91,92,93]. Afterwards, the samples were put into an oven (SnolTherm, UAB, Utena, Lithuania) with an Omron E5CC digital controller (OMRON ELECTRONICS GmbH, Langenfeld, Germany). The samples were annealed at temperatures ranging from 400 °C to 800 °C. Once the targeted temperature was reached using the maximum heating rate, the oven remained at the annealing temperature for 3 h. Samples were then naturally cooled down to room temperature. For each of the annealing temperatures, at least seven samples were prepared to ensure the reproducibility of the results.

## 3. Experimental Methods

To study the properties of the TiO_2_ thin films that are related to their photocatalytic performance, we used various characterization methods. First, spectroscopic ellipsometry (SE) (Lincoln, NE, USA) was used to provide the thin film thickness and the optical properties, namely the refractive index and extinction coefficient with the latter directly related to the light absorption in the thin films. Secondly, atomic force microscopy (AFM) was used to study the thin film morphology, roughness, and grain size. Finally, the phase composition of the thin films was determined using X-ray diffractometry (XRD) and UV Raman spectroscopy.

### 3.1. Spectroscopic Ellipsometry (SE)

The SE measurements were performed using an M-2000 ellipsometer from J.A. Woollam CO., (Lincoln, NE, USA). The spectra were taken in the range of 0.7 eV to 5 eV with angles of incidence from 45° to 75° with a step of 5°. The acquisition time for each spectrum was 20 s. A focused beam probe was used to achieve a better lateral resolution of approximately 200 μm. Four samples were randomly chosen from each sample batch with the same annealing temperature for the multi-sample analysis. Each sample was measured once in the middle of the substrate. CompleteEASE software https://www.jawoollam.com/ellipsometry-software/completeease, accessed on 30 July 2023) was used for data processing.

### 3.2. Atomic Force Microscopy (AFM)

Atomic force microscopy images were obtained via a HORIBA XploRa AFM–Raman system (HORIBA Jobin Yvon GmbH, Bensheim, Germany). A NT-MDT NSG10 tip was used with its nominal tip radius of 10 nm. The tapping mode was used, and the scanned area was 2 × 2 µm^2^ with a scanning step size along the x and y directions of 4 nm. The surface roughness was determined using the root-mean-square (RMS) value of the AFM images in Gwyddion [94]. For the quantitative grain size analysis, we used the Watershed segregation algorithms, which consist of two steps. The first step is called the grain location phase, where a local minimum is addressed, and the second step is called the segmentation phase, where the grain boundaries are finally defined by ‘filling’ the valleys using ‘drops of water’ based on several criteria [95,96]. An optimized combination of tuning parameters was found and performed for all images when processing the AFM images.

### 3.3. X-ray Diffractometry (XRD)

XRD experiments were performed using a SmartLab diffractometer Rigaku (Europe SE, Neu-Isenburg, Germany), equipped with a rotating Cu anode operating at 9 kW. XRD in 2θ/ω geometry was performed to probe the crystallization and crystallite size using a parallel beam. The spectra were measured in the range from 20° to 50° and in the range of 65° to 72° with a step of 0.02°. The latter scanned range was chosen for the silicon (004) peak at around 69°. This peak was then considered as the internal reference, and all spectra were normalized to the peak intensity.

### 3.4. UV Raman Spectroscopy

UV Raman spectroscopy was conducted using a HORIBA LabRAM HR Evolution system (HORIBA Jobin Yvon GmbH, Bensheim, Germany) equipped with a Symphony CCD detector. A UV-adapted 40× objective was used. The excitation wavelength was 325 nm with a power of around 1 mW. A grating of 2400 L/mm and an acquisition time of 30 s were used. The Raman spectra were obtained at three random points on each sample in a back-scattered geometry. The three spectra were then averaged for further data analysis. The spectral resolution was around 5.6 cm^−1^, as measured from the width of the Rayleigh peak.

### 3.5. In Situ Fourier Transform Infrared Spectroscopy (FT-IR)

The in situ experiment was performed using a Bruker Vertex 80v FT-IR spectrometer (Bruker Optics GmbH & Co. KG, Leipzig, Germany) with a home-built photocatalytic gas (reactant)–solid (photocatalyst) reaction chamber. A Globar providing light in the infrared range from 400 cm^−1^ to 7500 cm^−1^ was used. A Deuterated Lanthanum α Alanine-doped TriGlycine Sulphate (DLaTGS) detector was used for light detection. A KBr beam splitter and an 8 mm aperture were used in the optical path. The scanner mirror velocity was set at 5 kHz. The transmission geometry was used. Each spectrum was acquired for 5 min (corresponding to 178 scans) to maintain a good signal-to-noise ratio.

## 4. Quantitative Analysis of the Photocatalytic TiO_2_ Thin Films

### 4.1. Optical Properties and Film Thickness

The thickness of the TiO_2_ thin films as a function of the annealing temperature is shown in Figure 1. The layer structure of the optical model used in the fitting procedure of the SE data is illustrated in the inset of Figure 1. The ellipsometry data from reference [97] are utilized for the silicon substrate with a native oxide layer. The optical response of the TiO_2_ thin films is represented by a Cody–Lorentz function [98,99,100,101]. The parameters presented in italic in the inset indicate the quantities obtained by the simulations.

The final thickness was taken as the average thickness of the TiO_2_ thin films annealed at the same temperature. The error bars correspond to the statistical standard deviations. There is a clear trend that the thickness of the TiO_2_ thin films decreased from 140 nm to about 80 nm with an increasing annealing temperature from 400 °C to 600 °C. For higher annealing temperatures, the film thickness remained almost constant at around 80 nm. The decrease in the thickness was due to the crystal phase change and the densification of the thin films with an increasing annealing temperature [102,103,104]. When the samples were annealed at 800 °C, the thickness of the SiO_2_ also increased from 1–2 nm (native oxide) to approximately 55 nm. During the annealing process, oxygen molecules diffused through the film surface and reacted with silicon at the interface between the thin film and the substrate, while the silicon atoms diffused into the TiO_2_ thin film, thus forming oxide layers with larger thicknesses [105,106].

Figure 2 shows a representative example of the measured (open circles) and fitted results (dash-dotted line) Ψ and Δ spectra of the TiO_2_ thin films at an angle of incidence of 70°. Ψ (Psi), referring to the amplitude ratio of the complex reflection coefficients for p-polarized and s-polarized light, provides information about the magnitude of the reflected elliptically polarized light. Δ (Delta) represents the phase difference between p-polarized and s-polarized light reflected by the sample. Both Ψ and Δ depend on the film thickness and refractive index. The surface roughness affects Ψ more, while, e.g., anisotropy has a more pronounced effect on Δ.

The refractive indices, extinction coefficients, and optical bandgaps of the TiO_2_ thin films annealed at different temperatures are shown in Figure 3. The optical bandgaps were derived by taking the intercept on the abscissa of a linear fit of $\sqrt{\frac{\alpha }{h\nu }}$, as proposed by Cody et al. [107]. α represents the absorption coefficient, which was calculated using the extinction coefficient, k, using the formula $\alpha =\frac{4\pi k}{\lambda }$.

A trend for an increasing refractive index with ascending annealing temperature was observed. The increase in the refractive index indicates the densification of the thin films and agrees with the fact that rutile appears to be optically denser than anatase [108,109]. The optical bandgaps experienced a slight increase from 3.36 eV to 3.45 eV for temperatures ranging from 400 °C to 520 °C due to the further crystallization of the anatase phase, thus showing higher values. Further increasing the annealing temperature, which produced the phase transformation from anatase to rutile, led to a decrease in the optical bandgap from 3.45 eV to 3.28 eV, in agreement with the smaller bandgap of rutile [68]. However, a lack of consistency was observed regarding the optical bandgap of anatase and rutile in the literature mentioned above, which is due to the fact that TiO_2_ samples can be prepared using various methods and any small variation in the sample preparation process can influence the final properties of the material. Such inconsistencies are expected, and one should thus always be careful when comparing one’s own data with values in the literature.

A pronounced decrease in the optical bandgap was observed from around 3.28 eV to 3.13 eV when the annealing temperature reached 800 °C. This could be due to contamination from the fused silica, which was responsible for the lattice defects, thus reducing the bandgap [106] or the growth of the SiO_x_ layer at the film–substrate interface, the intermixing of the TiO_x_ and SiO_x_ phases, and the possible formation of Ti-O-Si bonds, which all together can contribute to the lowering of the values of the dielectric constants, as described in Ref. [105]. Our results clearly demonstrate the dependence of the optical bandgap on the phase composition.

### 4.2. Surface Morphology

The surface properties, such as the roughness and grain size of thin films are believed to be essential for their photocatalytic performance. AFM images of the TiO_2_ thin films annealed at different temperatures are shown in Figure 4.

#### 4.2.1. Surface Roughness

The AFM roughness is plotted in Figure 5, together with the roughness obtained using SE. An increase in the AFM roughness with ascending annealing temperatures was observed from around 0.8 nm at 400 °C to about 6.5 nm at 800 °C, as shown in Figure 5. This increase in roughness was predominantly due to the presence of the rutile phase when the thin films were annealed at higher temperatures. In the annealing temperature range from 400 °C to 520 °C, the variation in the roughness remained low at around 1 nm, which was almost at the limit of the AFM resolution. This ‘flat’ surface was directly correlated with the presence of the anatase phase. Within the annealing temperature range from 540 °C to 660 °C, the roughness increased almost linearly up to a value of around 5.5 nm. Within this range, we observed the presence of both anatase and rutile phases, and the ratio of rutile to anatase phase also increased as a function of the annealing temperature, as shown by both UV Raman spectroscopy and XRD (see later). Notice that only rutile was detected above the annealing temperature of 640 °C, and in the temperature range from 680 °C to 780 °C, the roughness increased further but not as fast as in the previous region. This is not surprising, because the dramatic change in the surface roughness was correlated with a phase change that is clearly observed in Figure 4. Since the rutile phase is a more thermally stable phase compared to anatase, we therefore did not expect a huge variation in surface roughness. The stronger increase in roughness at 800 °C was most likely due to intermixing with the silicon substrate, which influenced the surface morphology drastically [106].

The same trend in the surface roughness was observed using SE. The SE roughness was calculated based on the optical model, which included a roughness layer using the Bruggeman effective media approximation (EMA) [110]. This EMA approach simulates roughness as a layer consisting of a material-to-void ratio of 50% to 50%. Such an estimation usually performs well when the size of the surface roughness is much less than the wavelength of the light used for the analysis [111]. Usually, the surface roughness measured by AFM is less than the roughness measured by SE [112,113,114]. Our results (Figure 5) confirm this fact, especially in the annealing temperature range from 520 °C to 800 °C. There is an exception when the roughness is very low, as observed for the temperature range between 400 °C and 520 °C. In this range, adding the roughness layer in the SE data analysis as a fitting parameter did not produce any improvement compared with the model where a roughness layer was not considered. Therefore, modeling was performed considering zero roughness as a robust estimation for the real structure of the thin film samples. We can see that the AFM roughness increased almost linearly, especially in the range where the phase transfer occurred from 540 °C to 640 °C, while the SE roughness did not exactly follow a same trend. This difference could have originated from the approaches used to obtain these two types of roughness. The AFM roughness was directly measured on the surface and was thus more sensitive to any phase-transformation-induced change in the morphology. Such phase transformation is confirmed by other characteristic methods. The SE roughness, however, was simulated indirectly via an optical model and thus was influenced by the surface optical response to the polarization. This simulation depended not only on its morphology, but also on the optical properties related to the crystal structure and thin film compactness.

#### 4.2.2. Grain Size

From Figure 4, we can clearly see that the grain sizes varied with increasing annealing temperatures. Statistical histograms of the grain sizes represented by the projected boundary lengths are shown in Figure 6 [95,115,116]. A Gaussian function was used to fit the histograms [117]. The insets presented in Figure 6 show the segregation results, where the definition of grains on the surface of the TiO_2_ thin films is represented by the red mask in each AFM image. The boundary length of each segregated area (mask in red) was considered to be the grain size for each of the samples.

Statistical and quantitative analyses of the grain size and grain size distribution are indicated by the centroid position and the FWHM of the Gaussian profiles presented in Figure 7. The error bars were derived from the fitting process.

When the samples were annealed at 400 °C, it was very difficult to define the ‘grains’, because the surfaces of the samples were very smooth, making the segregation process difficult. Nevertheless, we still used the optimized image processing parameters of the AFM images to estimate the grain size. The highest statistical value corresponded to approximately 45 nm, which was most likely a result of the combination of several individual nanograins, since their boundaries could not be defined efficiently. However, this inversely proves that the surface was the smoothest among all samples. Starting from an annealing temperature of 450 °C, where pure anatase existed, to 640 °C, where the anatase phase completely disappeared, there was always a strong statistical count at a grain size of around 10 nm below the center of the Gaussian fitting. This single statistic value was not included in the fitting curve, and its existence was probably due to the presence of nanocrystalline anatase. Once the anatase phase was not observed any more, this peak was no longer significant, as shown by the grain size distribution of the annealing temperatures from 640 °C to 800 °C.

The center of the Gaussian profile indicates the mean grain size for a specific sample, while the width indicates the distribution of the grain sizes. We clearly saw a slow increase in the average grain size from around 45 nm to 100 nm within the annealing temperature range from 450 °C to 620 °C. Before 540 °C, where rutile appeared, such an increase was due to the crystallization of small nanocrystalline anatase. From 540 °C to 620 °C, the phase transfer took place and introduced a steady increase in the grain sizes. At 640 °C, where the phase transfer was completed, the grain size experienced a dramatic increase up to 680 °C. This indicates the formation of larger grains of rutile from small nanocrystalline rutile grains. Above 680 °C, the grain size only increased by around 25 nm at 780 °C. This small increase corresponds to further crystallization of the rutile phase. At 800 °C, the grain size tended to drop again. This phenomenon can be explained by the interaction between the TiO_2_ thin film and the Si substrate, where the thickness of SiO_2_ increased drastically, thus changing the morphology of the TiO_2_ thin film on top of it.

Besides the average grain sizes of the TiO_2_ thin films, the grain size distribution also varied with the annealing temperatures. From 400 °C to 500 °C, the width of the Gaussian distribution slightly decreased from around 200 nm to 100 nm, which means that anatase nanocrystallites combined and formed larger grains with a narrower distribution. From 500 °C to 620 °C, the width of the Gaussian distribution almost remained constant at 120 nm, but the corresponding physical processes behind it may have been quite different. We already know that, from 500 °C to 540 °C, pure anatase exists, while from 540 °C to 620 °C, anatase and rutile coexist. Considering that the average grain size within this range gradually increased while the distribution remained constant, this process could correspond to a uniformity of the growth of anatase grains and the following phase transformation. From 620 °C to 700 °C, where the phase transformation was first completed at 640 °C and then pure rutile existed, a dramatic increase in the Gaussian profile width was observed. This could indicate that some small rutile nanocrystallites started to grow at specific positions on the thin film, while other small rutile nanocrystallites could not form larger grains yet, thus creating a larger distribution. From 720 °C to 780 °C, the width of the grain size profile decreased and remained stable. This means that small rutile nanocrystallites finally formed grains with an average size of 240 nm without changing its distribution. The overall trend confirms the mechanism of slow rutile nucleation and fast growth [118].

### 4.3. Crystal Structure and Phase Analysis

The XRD diagrams presented in Figure 8 reveal that the films were composed of polycrystalline TiO_2_ with the following sequence with respect to the crystalline phases: (1) formation of the polycrystalline anatase crystal structure with (101), (200), (112), (004), and (103) reflections in the annealing range from 400 °C to 520 °C; (2) the coexistence of anatase and rutile in the range from 540 °C to 640 °C; and (3) polycrystalline rutile with a dominant (110) reflection and weaker reflections at (112), (111), and (219) from 660 °C to 800 °C.

The kinetics of the phase transformation from anatase to rutile are described in Ref. [118]. Anatase {112} twins develop through oriented attachment, and these twins play a crucial role in initiating the anatase-to-rutile phase transformation. Because rutile particles retain the same anatase twin morphology, there could be a crystallographic relationship between the two polymorphs by inferring the pre-transformation orientation of anatase, together with atomic displacements, which are necessary for the anatase to rutile phase transformation [119]. Therefore, the existence of the anatase (112) reflection in Figure 8 (right) is a good indicator of a successful phase transformation from anatase to rutile when thermal energy was introduced by annealing. At the same time, the phase transformation is also related to the anatase grain size [120,121,122], which influences the rate of anatase grain size growth and the phase transformation.

A quantitative analysis of the phase crystallinity was performed by employing the peak area and the full width at half maximum (FWHM) of the representative peaks of anatase at 25.35° and rutile at 27.47° in Figure 9 [123,124]. In the annealing temperature range from 400 °C to 520 °C, where only the anatase phase was found, the peak area remained almost constant while the FWHM decreased. This is attributed to the improved crystallinity of anatase when exposed to higher annealing temperatures. In the annealing temperature range from 540 °C to 640 °C, where both the anatase and rutile phases exist, we saw an increase in the peak area representing rutile and a decrease in the peak area representing anatase. This indicates that the ratio of anatase to rutile inside the thin film changed with the annealing temperature. The FWHM, on the other hand, decreased for both phases in this temperature range. The higher FWHM value for anatase, together with its lower peak area at 640 °C, indicate that the anatase crystallites deteriorated in crystallinity, while ordered rutile crystallites were formed. In addition, the error of the fitting for the FWHM dramatically increased at this temperature. In the annealing temperature range from 660 °C to 800 °C, the peak area representing rutile remained almost constant, while the FWHM slightly decreased with an ascending annealing temperature, indicating a small change in the amount of rutile with slightly increased crystallinity.

The Scherrer equation is widely used to estimate the crystallite size of polycrystalline samples in XRD quantitative analyses [125]. This equation relates the FWHM of the observed XRD peaks with the crystallite size, $FWHM=\frac{k\times \lambda }{\begin{array}{c} D\times \cos\left(\theta \right) \end{array}}$, where k is the shape constant, D is the crystallite size, λ is the used excitation wavelength, and θ is the Bragg angle in radians. Despite the limitations of using Scherrer equations [126], particularly with respect to the instrumental broadening, strains, and/or disorders in the sample that could contribute to the diffraction peak [127], the robustness of using Scherrer formulas with different crystal sizes, and the number of diffraction peaks in a XRD diagram that need to be used for calculating the crystallite size [128], we applied the Scherrer equation to estimate the crystallite sizes of anatase and rutile, as this was shown to provide relatively reliable results [129]. The characteristic features of anatase at around 25° and rutile at around 27° were used without taking the strain effects into consideration. Nevertheless, it is important to keep this in mind when interpreting the result showing that omitting such an effect could lead to an underestimation of the size of the coherent diffraction domain [127].

The results are shown in Figure 9 (right). The crystallite size increased from around 16 nm to 25 nm for annealing temperatures from 400 °C to 520 °C, which indicates improved crystallinity of the anatase phase with an increasing annealing temperature. In Figure 7, the grain size at the surface obtained from the AFM analysis also increased from around 45 nm to 75 nm. Considering that we used the projected boundary length as an estimation of the grain size, which is basically the perimeter of an arbitrary shape, we can roughly describe the grains geometrically as a circle with the same circumference. Therefore, the equivalent circles with circumferences of 45 nm and 75 nm correspond to circles with diameters of approximately 14 nm and 24 nm, respectively. These values are in the same range of crystallite sizes as those derived from the Scherrer equation. In the annealing temperature range from 540 °C to 620 °C, where both anatase and rutile existed, and keeping in mind that the anatase phase was hardly detectable at 640 °C, the anatase crystallite size increased from around 22 nm to 29 nm, while the rutile crystallite size increased from around 28 nm to 33 nm. Using a similar geometrical approximation as before, we obtained grain sizes from 73 nm to 96 nm that corresponded to equivalent circle diameters from 23 nm to 31 nm. This is very interesting because for an annealing temperature of 540 °C, the average grain size (23 nm) was more defined by the anatase crystallite size (22 nm), while at 620 °C, the AFM grain size of 31 nm coincided with the average value of the XRD derived crystallite sizes of 29 nm and 33 nm for the anatase and rutile phases, which made it difficult to simply compare numbers to decide which phase dominated more at this annealing temperature. However, the estimated anatase crystallite size at 620 °C of 29 nm in Figure 9 (right) seems to be out of scope, and we, therefore, believe that at 620 °C, the dominant factor influencing the average grain size was the rutile phase with a crystallite size of 33 nm. In the annealing temperature range from 640 °C to 800 °C, there was a continuous increase in the crystallite size from 33 nm to 36 nm, while the equivalent diameter was from 41 nm to 73 nm, as derived from grain sizes from 130 nm to 230 nm. This deviation is mostly due to the grains containing more than one crystallite [130].

Furthermore, UV Raman spectra with an excitation wavelength of 325 nm were also obtained to study the phase and crystal structure. The results were thereafter compared with the XRD results. The representative Raman spectra are shown in Figure 10. The red/blue dot-dashed lines presented in Figure 10 correspond to the characteristic frequency positions of anatase/rutile vibrational modes.

The Raman spectra of rutile, which belong to the space group $D_{4h}^{14}-P4_{2}/mnm$, have four Raman active vibrations, namely one B_1g_ mode at 145 cm^−1^, one two-phonon scattering mode at 235 cm^−1^, one E_g_ mode at 447 cm^−1^, and one A_1g_ mode at 612 cm^−1^. Anatase, which belongs to the space group $D_{4h}^{19}-{I4}_{1}/amd$, has six Raman active vibrations, including two E_g_ modes at 144 cm^−1^ and 197 cm^−1^, a B_1g_ mode at 399 cm^−1^, an A_1g_ mode at 515 cm^−1^, and another B_1g_ mode at 519 cm^−1^, as well as an E_g_ mode at 638 cm^−1^ [131,132].

There was a slight peak shift to lower wavenumber for anatase modes at around 400 cm^−1^ and 198 cm^−1^ from 400 °C to 640°C, which was due to compressive stress in the annealed samples. The variation in the UV Raman spectra reveals the phase transformation with the annealing temperature. This became very clear in the spectral range of about 600–650 cm^−1^, where a gradual change from the anatase peak at 638 cm^−1^ to the rutile peak at 612 cm^−1^ was apparent. This agrees nicely with the XRD results and confirms that a mixture of anatase and rutile was present in the annealing temperature range from 540 °C to 640 °C, while below and above this range, only anatase and rutile existed, respectively.

To quantify the contributions of the crystalline phases, a peak analysis by fitting of the representative features for anatase and rutile was performed. The challenge of choosing the appropriate characteristic peaks for anatase and rutile still remains, because many physical parameters can influence the shape and intensity of the Raman peaks in the TiO_2_ samples, such as the particle size, crystallinity, and impurities [133,134,135]. For instance, if the crystallite size decreases below the phonon mean free path, the phonon peaks slightly shift, broaden, and become asymmetric, and the intensity decreases [136]. For anatase, the prominent peak at 144 cm^−1^ was not chosen for the data analysis, because the edge filter in the spectrometer resulted in a cut-off at around 100 cm^−1^ with a tail. This thus created difficulties and uncertainties in the peak fitting process. The peak at around 197 cm^−1^ was also not chosen because of its low signal intensity. The peak doublet of 515 cm^−1^ and 519 cm^−1^ was also not used in the analysis, as a minor contribution from the silicon substrate, which typically appears at 520.7 cm^−1^, could not be excluded, even though the light was strongly absorbed in the TiO_2_ films [137]. Therefore, finally, the two anatase modes at 399 cm^−1^ and 638 cm^−1^ as well as the rutile one at 612 cm^−1^ were chosen for the peak analysis. Voigt functions were used for peak fitting. The Voigt function as a convolution of Gaussian and Lorentzian functions can incorporate Gaussian contributions not only through instrumental broadening, but also through inhomogeneous broadening, e.g., by the nanocrystalline size distribution. The experimental resolution was derived from the measurement of the width of the Rayleigh line (elastically scattered light), which was 5.6 cm^−1^ in this case. We therefore set a minimum value of 5.6 cm^−1^ for the Gaussian width when performing the peak fitting. The FWHMs of the Voigt function were finally used for comparisons among all samples, since the values were considerably larger than the experimental broadening. The integrated peak areas under the fitted Voigt function and the widths of the Voigt function for anatase and rutile are plotted in Figure 11.

The intensities (peak areas) of the anatase modes at both 638 cm^−1^ and 399 cm^−1^ hardly changed for annealing temperatures from 400 °C to 560 °C and then started to decrease. The rutile mode at 612 cm^−1^, on the other hand, revealed a continuous increase in the intensity from 540 °C to 680 °C and then remained constant up to 800 °C. Consistent with the XRD results, a mixture of anatase and rutile phases was present in the annealing temperature range from 540 °C to 640 °C.

The FWHMs of the anatase peaks at 399 cm^−1^ and 638 cm^−1^ both decreased with ascending annealing temperatures from 400 °C to 640 °C. The FWHM of the rutile peak at 612 cm^−1^ increased for temperatures from 540 °C to 680 °C and then remained approximately constant up to 800 °C. This behavior was unexpected, taking into account the XRD results, which showed decreasing FWHMs with increasing temperatures, thus indicating increasing crystallite sizes. The latter should have, in general, resulted in a lower Raman FWHM as well. The evolution towards very large Raman FWHM values of about 47 cm^−1^ with the annealing temperature thus requires further attention.

A comparison of the ratios of phases using the peak areas from the UV Raman and XRD results defined by $\frac{{PA}_{anatase\;or\;rutile}}{{PA}_{anatase}+{PA}_{rutile}}$ is shown in Figure 11. A linear fit, y=ax+b, was used, and the parameters of the slope and the intercept are summarized in Table 1.

The closeness of parameters a and b derived from both the Raman and XRD measurements provide us with confidence in the determination of the phase ratio and its evolution with the annealing temperature.

## 5. Photocatalytic Activity

To test the photocatalytic properties of the TiO_2_ thin films annealed at temperatures from 400 °C to 800 °C, a self-designed photocatalytic reaction chamber was developed, which was then used in combination with in-situ Fourier transform infrared spectroscopy (FTIR).

The photocatalytic reaction that was chosen for the evaluation of the photocatalytic properties of the various TiO_2_ thin films was the photodegradation of acetone gas into carbon dioxide (CO_2_) following the typical chemical reaction pathway, as described in Ref. [138].

The TiO_2_ photocatalytic thin films annealed at various temperatures were placed in the chamber. They were illuminated using a commercial LED chip with an excitation wavelength of 365 nm and a radiation power density of around 10 mW/cm^2^ at the sample position. A total of 100 mL of saturated acetone gas from an acetone reservoir at atmospheric pressure and temperature was injected into the reaction chamber to produce a saturated amount of reactant acetone in the reaction chamber.

Ten minutes after the injection of the acetone gas, we started to record the transmission spectra of the gas composition inside the reactor. The first spectrum was always taken without UV light irradiation; therefore, the spectra indicate that the initial gas mixture was mainly composed of acetone. Once the first spectrum acquisition had finished, the LED was switched on immediately and the next 24 spectra were continuously recorded within the next 2 h.

Figure 12 demonstrates the typical IR transmission spectra of the gas mixture in the reaction chamber from 0 min to 120 min using the TiO_2_ thin film annealed at 500 °C. A marked increase in the amount of CO_2_ inside the chamber was observed by inspecting the characteristic IR features of CO_2._ The two modes that were assigned to the CO_2_ gas at around 668 cm^−1^ and 2347 cm^−1^ (red bars in Figure 12) correspond to the bending mode and the asymmetric stretching of the CO_2_ molecule, respectively, and these became more visible in the transmission spectra with an increasing reaction time, indicating continuous photodegradation of the acetone gas [139].

A quantitative analysis was performed using the integration of the CO_2_ characteristic IR feature in the transmittance spectra from 2270 to 2400 cm^−1^. The integrated area was fixed for all spectra. The integrated area and the baseline are shown in Figure 13. The small amount of CO_2_ that was already present at 0 min is due to the injection of the saturated acetone gas from the reservoir, where some CO_2_ was present in the air. These integrated values are a straightforward indicator of the amount of CO_2_ in the reaction chamber. Therefore, the difference in such values was then used to represent the effectiveness of the photocatalytic process for TiO_2_ thin films annealed at different temperatures.

It is important to note that the amount of CO_2_ is directly related to the amount of photocatalyst used. Here, we needed to define the microscopic area of the TiO_2_ thin films to perform a proper comparison among them afterwards. We therefore employed an image processing algorithm in ImageJ [140] that was based on the color difference between the area covered by TiO_2_ thin film and the pure silicon substrate (see the Appendix A). The integrated areas for CO_2_ from the IR transmission spectra represented by different TiO_2_ photocatalytic thin films were then normalized to the derived area of the TiO_2_ thin films. The photocatalytic performance was compared using the normalized area of CO_2_ (A_CO2_), as presented in Figure 14 (left). All photocatalytic measurements were performed multiple times, and a statistical error bar was created.

Figure 14 (left) demonstrates the pronounced differences in the photocatalytic performance of the TiO_2_ thin films that were annealed from 400 °C to 800 °C. Within two hours, the amount of CO_2_ that was produced via the photodecomposition of acetone gas gradually increased. We therefore took the value at 120 min for each sample as the indicator of the photocatalytic efficiency. Figure 14 (right) shows that the best photocatalytic performance was provided by the TiO_2_ samples annealed at 500 °C. For samples annealed at higher temperatures of above 500 °C, the photocatalytic capability decreased dramatically from 500 °C to 540 °C and then slowed down and stayed almost stable until 640 °C, which is interesting because the linear decrease in the ratio of the anatase phase in the thin films did not seem to influence the photocatalytic performance linearly as we might have expected. Further increasing the annealing temperature from 660 °C to 800 °C led to even worse photocatalytic performances. A similar photocatalytic performance trend was also observed in Ref. [141], where TiO_2_ thin films were prepared via spin coating with a different recipe and annealed in the temperature range from 200 °C to 1000 °C with much larger annealing temperature step sizes of 100 °C or 200 °C.

To understand why the TiO_2_ thin films annealed at 500 °C outperform other samples, we utilized a correlation matrix, as described in the next section, to link all of the characterized parameters regarding the thin film’s optical and structural properties as well as the morphology with the photocatalytic properties.

## 6. Correlation Matrix

Correlation analysis is a commonly used statistical method to study the relations among measured parameters in many fields of research [142,143,144] due to its ability to provide a constructive view of big data sets. However, many correlation coefficients have been proposed and we employed a thorough comparison and discussion in ref. [145]. Among them, three correlation coefficients are usually applied, namely the Pearson, Spearman, and Kendall coefficients. The robustness of such correlation coefficients is highly related to the number of samples. Nevertheless, these three methods generally offer similar results, with the Pearson coefficient being more sensitive to outliers, while the other two methods show similar performance levels in terms of correlation directions and significant relationships for the same sample sizes [146].

Based on this background knowledge, we quantified all of the properties obtained by the above-mentioned characterization techniques, including the TiO_2_ thin film thicknesses, refractive indices, and extinction coefficients at a wavelength of 365 nm by SE; the surface roughnesses, grain sizes, and grain size distributions by AFM; and the phase compositions and crystallinities obtained from XRD and UV Raman spectra. We put them into a final matrix using the Kendall coefficients, as shown in Figure 15. We produced the correlation matrix using the other two correlation coefficients [145,147], namely the Pearson and Spearman’s Rank. The results for the Spearman’s Rank were very similar to those of the Kendall’s Rank, and the results from the Pearson analysis were slightly different with specific coefficients. However, the color pattern in the matrix, which presents the direction of the correlation was the same. The difference could be due to the fact that Pearson correlation coefficients focus more on the linear relationship between two variables, and the other two coefficients focus more on the dependence of variables [146]. For simplicity, here, only the Kendall correlation matrix is shown.

In Figure 15, the color of the scale bar goes from −1 (blue) to 1 (red) with the plus/minus signs corresponding to the directions of the relations as positive/negative. The higher the absolute values of the coefficients, the more significantly the two variables depend on each other.

Keeping in mind that our initial objective was to study the dependence of the photocatalytic performance on the annealing temperature, we noticed that the coefficient between the annealing temperature and the CO_2_ production showed that increasing the annealing temperature hindered the photocatalytic performance of the TiO_2_ thin films. The annealing temperature had significantly positive correlations with the roughness (0.96), average grain size (0.91), refractive index (0.7), grain size distribution (0.57), and extinction coefficient (0.78) of the thin films. With respect to specific phases, the annealing temperature showed significant negative correlations with anatase-phase-related properties, for instance, the peak area for anatase determined by UV Raman spectroscopy (−0.76) and XRD (−0.86) and the crystallinity of the anatase phase indicated by the FWHM determined by UV Raman spectroscopy (−0.76) and XRD (−0.86). Conversely, the annealing temperature showed, in general, a strong positive correlation with rutile-related properties.

In the second line of the correlation matrix, there are five ‘red’ indicators that give positive evaluations of the photocatalytic performance, including the thickness of the TiO_2_ thin films, the peak area, and the FWHM for the representative peak of anatase measured by both XRD and UV Raman spectroscopy. Considering the absolute values of the correlation coefficients, the main reasons for the better photocatalytic properties of the TiO_2_ thin films may be the presence of the anatase phase and the thickness of the thin films. Note that all property aspects were correlated with each other, so we can conclude that between these two main aspects, a correlation coefficient of 0.49 for the thickness and a correlation coefficient of 0.85 for the crystallinity determined by XRD, the crystallinity is the one that seems to play a major critical role among these parameters [148,149].

It is commonly accepted that the photocatalytic performance is dependent on the surface area of the photocatalyst, especially for powders or mesoporous structures [150]. Similarly, we expect that a greater roughness in the thin film would dramatically influence the photocatalytic performance positively. However, in our samples, the surface roughness did not contribute to the photocatalytic performance in a positive way, as shown in all correlation matrices. This could be due to the fact that the roughness here was basically due to the different phase compositions of the TiO_2_ thin films. Compared to high-surface-area structures (e.g., porous structures), they can only be considered compact, dense, and “flat” thin films, which leads to no positive correlations. Furthermore, the negative relation between the roughness and the photocatalytic performance is most probably due to the phase composition difference, not to the roughness itself.

Finally, it is necessary to mention that this data analysis and conclusion are only applicable for this specific sample preparation process. Such a correlation matrix could be used to provide a prediction of the results by using this sample synthesis method, e.g., for other annealing temperatures, which are not included in this range.

## 7. Conclusions

In this study, we synthesized a set of TiO_2_ thin films on (100)-oriented p-type silicon substrates using a sol-gel process and a spin-coating deposition procedure. The as-prepared samples were annealed at different temperatures from 400 °C to 800 °C. The small temperature step size of 20 °C allowed us to systematically study the changes in the properties of the TiO_2_ thin film material itself brought about by the calcination temperature and how the relevant properties affected the thin film photocatalytic performance. The thin films were characterized using different techniques, including SE, XRD, AFM, and UV Raman spectroscopy, to determine their optical properties, thicknesses, phase compositions, and surface morphologies. The photodecomposition of acetone gas into CO_2_ was used as the test photocatalytic reaction to determine the photocatalytic performance of each TiO_2_ thin film. These properties were evaluated using a correlation matrix, and we found that the best active photocatalytic samples were the ones annealed at 500 °C with the responsible parameter being the existence of pure anatase with good crystallinity. Such a correlation matrix demonstrates a full view of all parameters that are relevant to this specific preparation method. The methodology of this study can be utilized as a template for systematic studies and analyses for any other sets of photocatalytic samples that may be influenced by the multiple properties of the sample and their preparation protocols.

## Figures and Tables

**Figure 1 materials-16-05494-f001:**
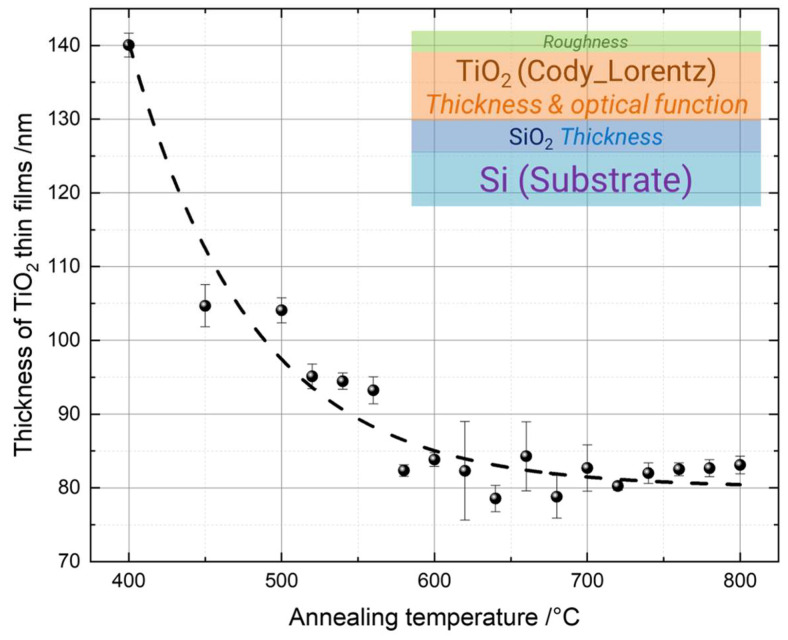
Thickness of TiO_2_ thin films annealed at different temperatures from 400 °C to 800 °C. Inset: layer structure of the optical model simulated in the SE data processing. The dashed line is used as a guide for the eye.

**Figure 2 materials-16-05494-f002:**
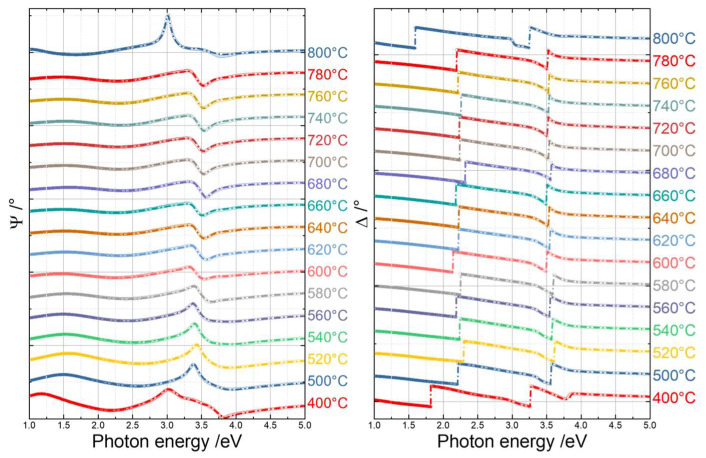
Experimental spectra (open circles) and fitted results (dash-dotted lines) of Ψ (**left**) and Δ (**right**) at an angle of incidence of 70°.

**Figure 3 materials-16-05494-f003:**
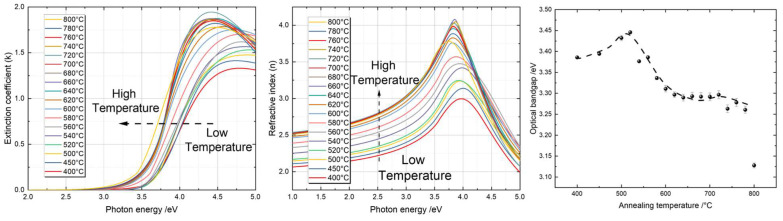
Refractive index (**left**), extinction coefficient (**middle**), and optical bandgaps (**right**) of the TiO_2_ thin films annealed at various temperatures. The dashed line connecting the optical bandgaps is a guide for the eyes.

**Figure 4 materials-16-05494-f004:**
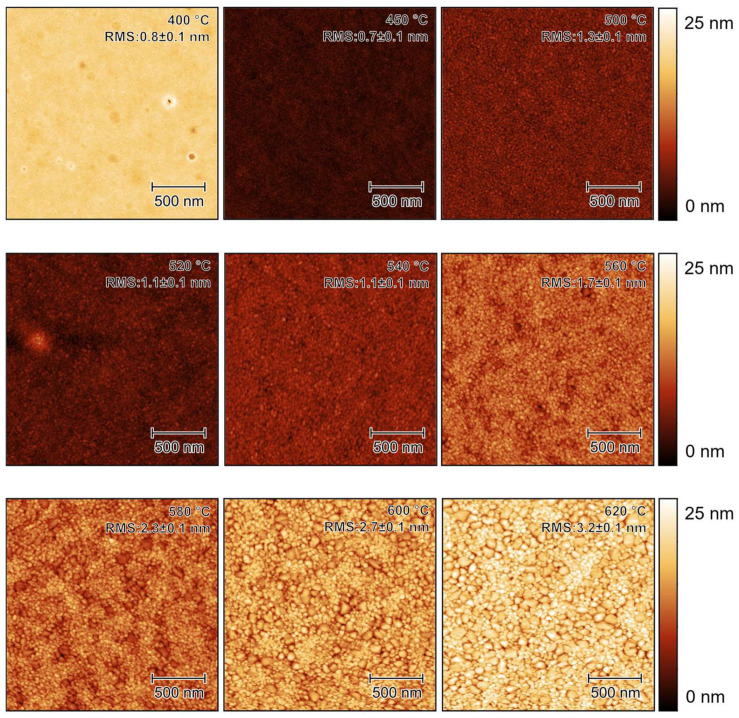
Typical AFM images of TiO_2_ thin films annealed at different temperatures. RMS roughness values are shown in individual images. For better visuality, different height scales are shared at the end of each line with the three images that are in the same line. The scale bar in all images is 500 nm.

**Figure 5 materials-16-05494-f005:**
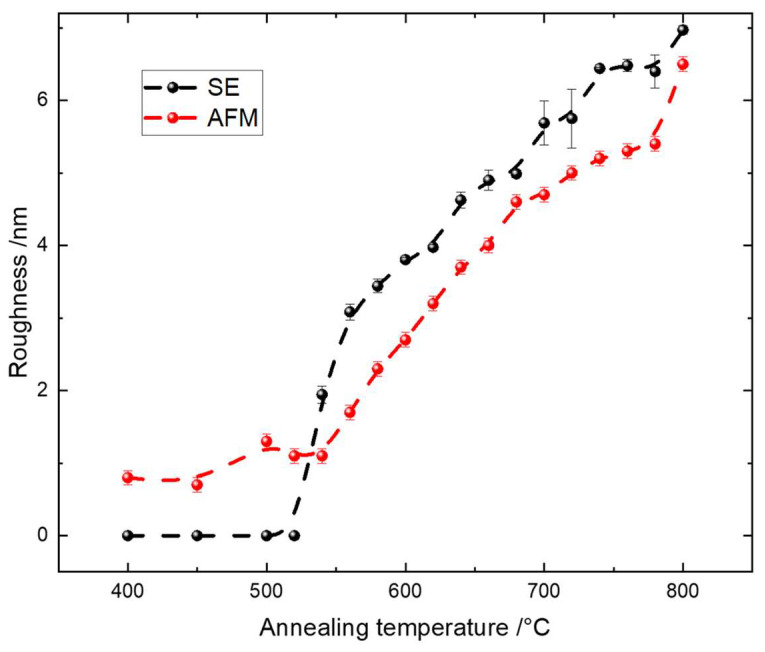
Roughness of the TiO_2_ thin films annealed at different temperatures using SE and AFM, where the dashed lines are guides for the eyes.

**Figure 6 materials-16-05494-f006:**
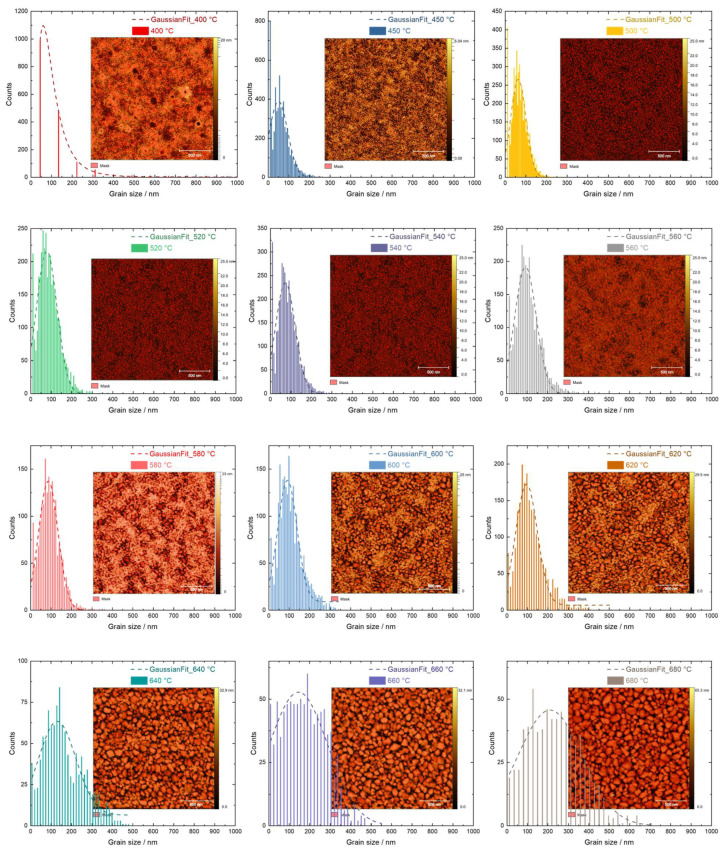
Histograms of the grain size distribution for the TiO_2_ thin films annealed at different temperatures together with a Gaussian fit of the profiles. The insets show the grains derived from the segregation method for each AFM image.

**Figure 7 materials-16-05494-f007:**
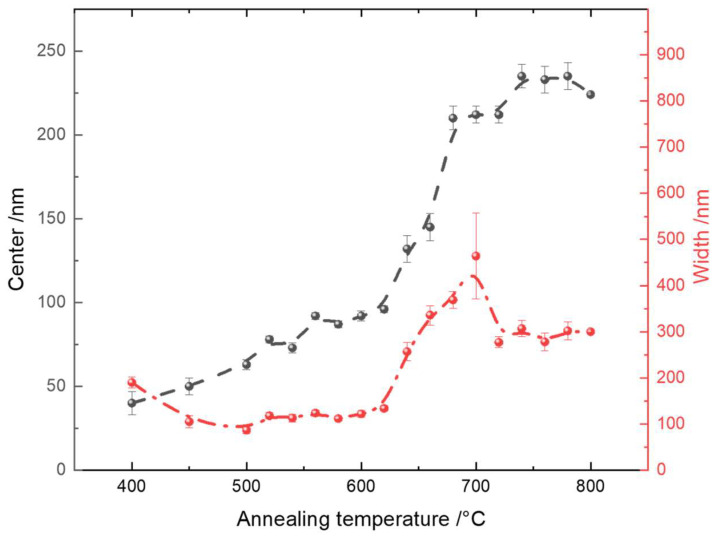
Grain size analysis represented by the center position and the width of the Gaussian fit from the histograms in Figure 6.

**Figure 8 materials-16-05494-f008:**
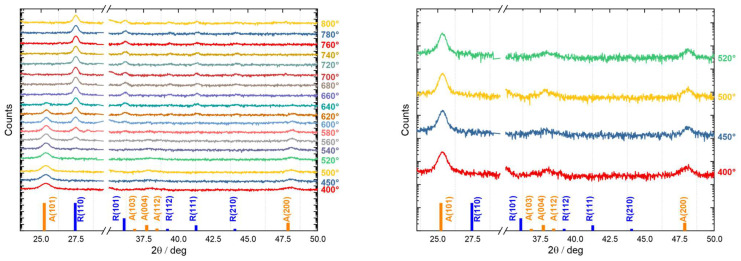
**Left**: XRD diagrams for TiO_2_ thin films annealed at different temperatures from 400 °C to 800 °C. **Right**: exemplary spectra indicating the presence of the anatase (112) reflection.

**Figure 9 materials-16-05494-f009:**
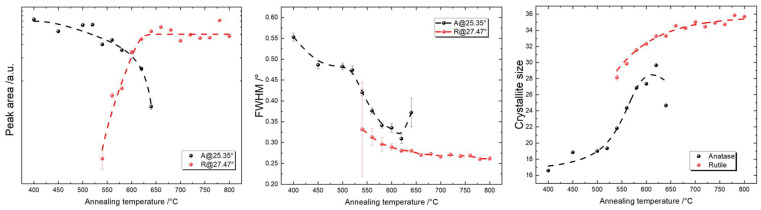
Peak analysis of XRD diffractograms with peak areas (**left**) and the FWHM (**middle**) of the representative peaks for anatase at 25.34° and rutile at 27.49°. **Right**: estimated crystallite sizes of anatase and rutile using the Scherrer formula.

**Figure 10 materials-16-05494-f010:**
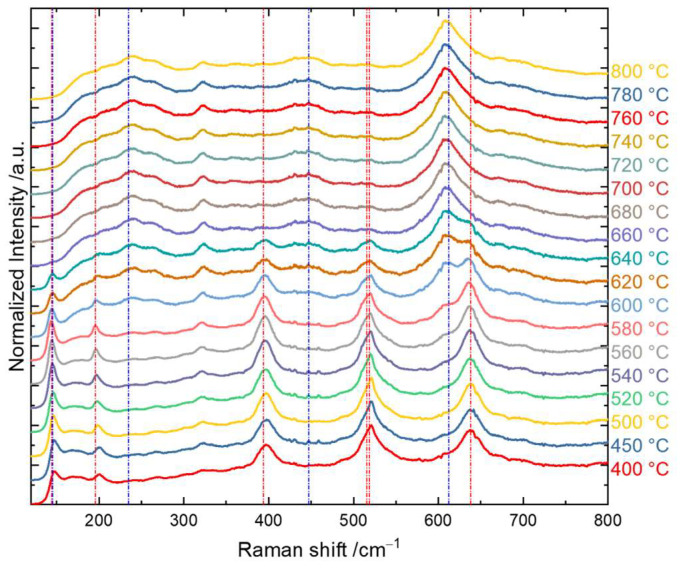
UV Raman spectra of each TiO_2_ thin film annealed at temperatures from 400 °C to 800 °C, where the red/blue dot-dashed lines correspond to the characteristic frequency positions of anatase/rutile vibrational modes.

**Figure 11 materials-16-05494-f011:**
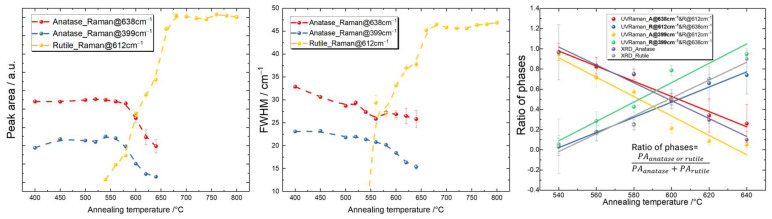
Peak area (**left**) and FWHM (**middle**) of the characteristic peaks of anatase and rutile at different annealing temperatures; the ratio of phases calculated using the Raman peak areas of different modes and compared to those obtained from the XRD data in the temperature range where anatase and rutile coexist (**right**).

**Figure 12 materials-16-05494-f012:**
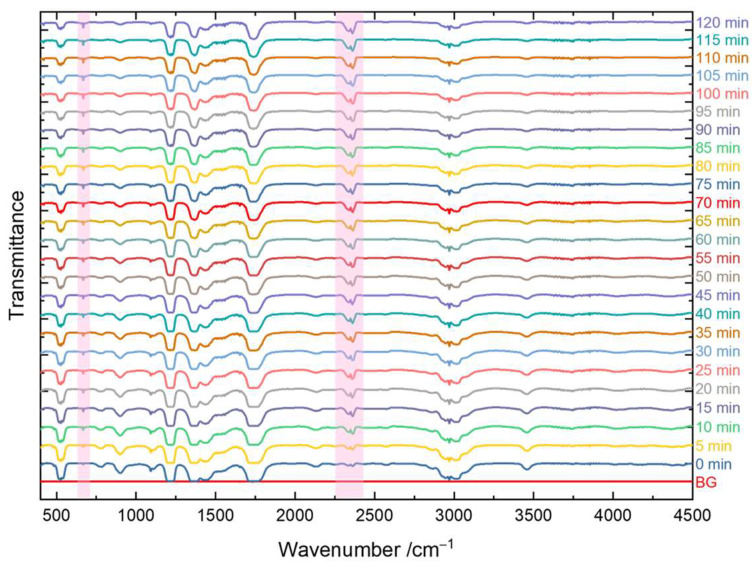
Typical IR transmittance spectra of the gas mixture in the reaction chamber during the 2 h of photodegradation of acetone gas using the photocatalytic TiO_2_ thin films annealed at 500 °C, where the red stripe area indicates the characteristic CO_2_ IR features.

**Figure 13 materials-16-05494-f013:**
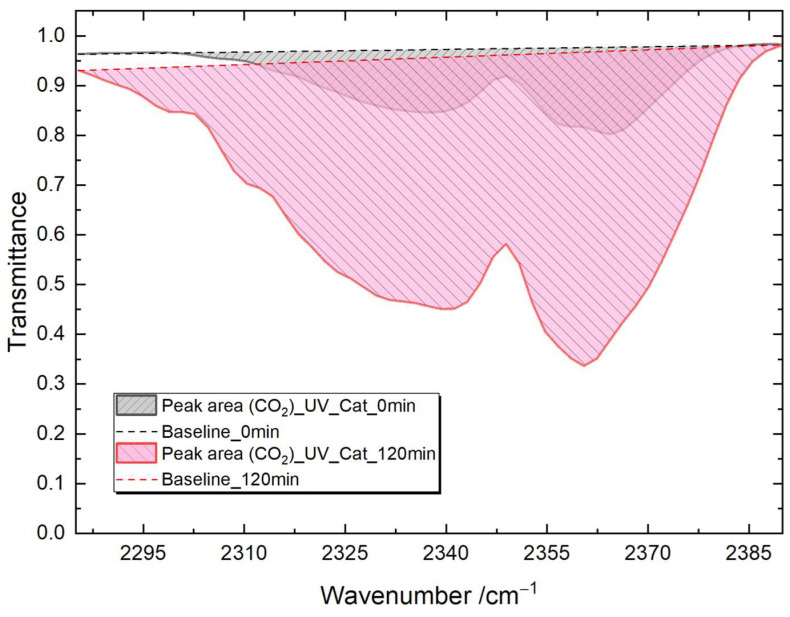
Data processing for the quantitative analysis on the characteristic CO_2_ feature with the baseline indicated by a dashed straight line and the integrated area as shaded region with tilted lines.

**Figure 14 materials-16-05494-f014:**
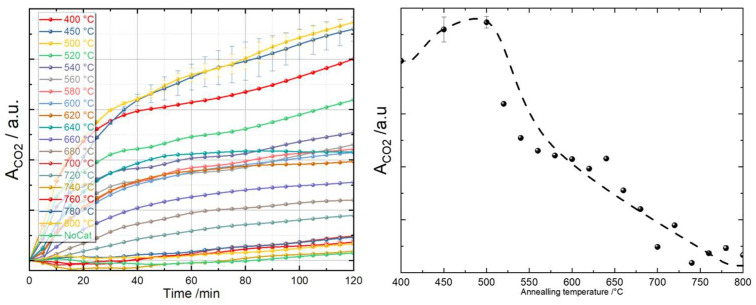
Production of CO_2_ normalized to the microscopic surface area of the TiO_2_ thin films annealed at different temperatures (**left**); the production of CO_2_ after 120 min using different TiO_2_ thin films, where the dashed line is used as a guide for the eyes (**right**).

**Figure 15 materials-16-05494-f015:**
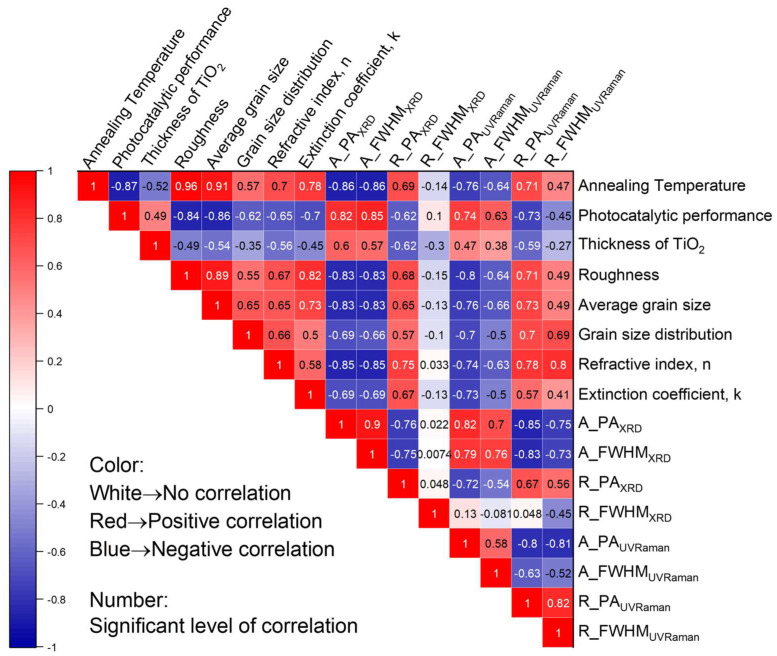
Correlation matrix with the quantified parameters including the annealing temperature, photocatalytic performance (CO_2_ production), thin film properties (thickness, roughness, average grain size, grain size distribution, refractive index, extinction coefficient, peak area, and FWHM for anatase and rutile obtained from XRD and UV Raman).

**Table 1 materials-16-05494-t001:** Summary of the linear fitting of the ratio of phases in Figure 11. (The text in black corresponds to the results analyzed using Raman spectroscopy, and the text in green corresponds to the results analyzed using XRD). The bold texts corresponds to the specific phase in the same line that is analyzed.

*y* = *ax* + *b*	*a*	*b*
Anatase@**638** cm^−1^ vs. 612 cm^−1^	−0.0075	5.0316
Rutile@638 cm^−1^ vs. **612** cm^−1^	0.0075	−4.0316
Anatase@**399** cm^−1^ vs. 612 cm^−1^	−0.0096	6.0835
Rutile@399 cm^−1^ vs. **612** cm^−1^	0.0096	−5.0935
Anatase@**25**° vs. 27°	−0.0088	5.7978
Rutile@25° vs. **27**°	0.0088	−4.7978

## Data Availability

The data presented in this study are available on request from the corresponding author.

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
