# Peer review of "Photocatalytic Performance of Sol-Gel Prepared TiO2 Thin Films Annealed at Various Temperatures"

_materials, 2023, doi:10.3390/ma16155494_

Round 1

Reviewer 1 Report

This study synthesized TiO2 thin films on p-type silicon substrates through a sol-gel process and annealed them at different temperatures. The films were characterized to determine their properties, and their photocatalytic performance was evaluated using acetone gas photodecomposition. The results showed that the best photocatalytic activity was observed in films annealed at 500 °C, where pure anatase phase with good crystallinity was present. The methodology employed in this study can serve as a guide for systematic analysis of photocatalytic samples influenced by multiple properties and preparation protocols. Here are some suggestions that may help improve the manuscript:

1-      Some sentences in the abstract are quite long and complex. Try breaking them down into shorter, more digestible sentences to improve readability.

2-      While it's good to provide an overview of the characterization techniques and experimental setup, consider including a brief mention of the specific results or findings obtained from the study in the abstract.

3-      While it's important to support your statements with references, the excessive use of citation numbers in the introduction can make the text appear cluttered. Consider integrating the citations more smoothly into the narrative or consolidating multiple references when possible.

4-      The discussion on the crystalline structures of TiO2 and their relevance to photocatalytic applications is informative.

5-      It would be beneficial to provide a brief explanation of why a combination of anatase and rutile phases might exhibit enhanced photocatalytic activity.

6-      The details of the material used (supplier, etc.) must be stated.

7-      How does the resistivity of the substrate impact the properties of the TiO2 thin films, especially in terms of their photocatalytic performance?

8-      What is the reason for using a two-step spin-coating process with different rotation speeds? How does this process help in achieving uniform and well-coated films?

9-      In XRD analysis, it would be helpful to discuss the relative proportions of anatase and rutile and their potential implications for the photocatalytic performance of the thin films.

10-   Could you provide possible explanations for the deteriorating crystallinity of anatase and the factors influencing the formation of ordered rutile crystallites?

11-   Refer to (DOI 10.1088/1361-6528/abfd54) and explain why they mention “the increase in the treatment temperature did not cause significant changes in the XRD results.” Do your results contradict this claim?

12-   Provide more details about the specific methodology used in the image processing algorithm.

13-   Investigate the stability and durability of the TiO2 thin films under prolonged photocatalytic reactions.

Overall, the English writing is clear and understandable.

Reviewer 2 Report

In this work, the authors demonstrated the growth of TiO2 films by sol-gel method and its photocatalytic performance as a function of annealing temperatures. However, the following points must be answered before acceptance for publication.

1. Abstract must be rewritten by emphasizing more on the novelty of this work.

2. There were few recently published works on TiO2 films prepared by the sol-gel method which can be referred to in the introduction part.

a. https://doi.org/10.3389/fmats.2023.1129886

b. https://doi.org/10.1007/s10971-021-05628-5

3. Whether Si substrate has a thermally grown oxide layer above it?

4. On page 5, line 188, authors stated ". When the samples are annealed at 800 °C, the thickness of the SiO2 also increases from 1-2 nm (native oxide) to approximately 55 nm." How did the authors measure the thickness?

5. In Figure 2 b, why do the samples anneal at 400 C have three features when compared to other annealing conditions?

6. On page 6, line 230, authors stated "Also, according to ref. [84], the growth of the SiOx layer at the film-substrate interface, the intermixing of TiOx and SiOx phase, and a possible formation of Ti-O-Si bonds, can contributes to lower values of the dielectric constants" Is it possible at 800 C since the temperature just increase by 20 C however bandgap changed drastically from 3.26 to 3.13 eV? 

7. Please do not use bullet points in between the text. Follow the standards of the journal. Rewrite them in different notation.

8. On page 11, line 371, the authors stated "Therefore, the existence of the anatase (112) reflection in Figure 8." However its very difficult to see anything in Figure 8. Pls, show an enlarged view of this region separately.

9. Although Raman showed some peak shift as a function of temperature, is there any shifts in the peaks from XRD results?

10. Why the error bar is very huge for TiO2 annealed at 540 C from Figure 9b?

11. In Figure 10, why the lines around 190, 390 and 520 are away from their peak position? Whether it should be Raman spectra or UV-Raman? 

12. Please make the units and font size the same for all the figures in the manuscript.

13. The authors stated that 500 C sample outperformed the photocatalytic performance when compared to others. Does this mean anatase phase is better than Other phases in Tio2 for photocatalytic studies?

Round 2

Reviewer 2 Report

Accept